# Learning Structured Output Representation using Deep Conditional Generative Models

**Kihyuk Sohn**[*†]     **Xinchen Yan**[†]     **Honglak Lee**[†]

[*] NEC Laboratories America, Inc.
[†] University of Michigan, Ann Arbor

ksohn@nec-labs.com, {xcyan,honglak}@umich.edu

## Abstract

Supervised deep learning has been successfully applied to many recognition problems. Although it can approximate a complex many-to-one function well when a large amount of training data is provided, it is still challenging to model complex structured output representations that effectively perform probabilistic inference and make diverse predictions. In this work, we develop a deep conditional generative model for structured output prediction using Gaussian latent variables. The model is trained efficiently in the framework of stochastic gradient variational Bayes, and allows for fast prediction using stochastic feed-forward inference. In addition, we provide novel strategies to build robust structured prediction algorithms, such as input noise-injection and multi-scale prediction objective at training. In experiments, we demonstrate the effectiveness of our proposed algorithm in comparison to the deterministic deep neural network counterparts in generating diverse but realistic structured output predictions using stochastic inference. Furthermore, the proposed training methods are complimentary, which leads to strong pixel-level object segmentation and semantic labeling performance on Caltech-UCSD Birds 200 and the subset of Labeled Faces in the Wild dataset.

## 1   Introduction

In structured output prediction, it is important to learn a model that can perform probabilistic inference and make diverse predictions. This is because we are not simply modeling a many-to-one function as in classification tasks, but we may need to model a mapping from single input to many possible outputs. Recently, the convolutional neural networks (CNNs) have been greatly successful for large-scale image classification tasks [17, 30, 27] and have also demonstrated promising results for structured prediction tasks (e.g., [4, 23, 22]). However, the CNNs are not suitable in modeling a distribution with multiple modes [32].

To address this problem, we propose novel deep conditional generative models (CGMs) for output representation learning and structured prediction. In other words, we model the distribution of high-dimensional output space as a generative model conditioned on the input observation. Building upon recent development in variational inference and learning of directed graphical models [16, 24, 15], we propose a conditional variational auto-encoder (CVAE). The CVAE is a conditional directed graphical model whose input observations modulate the prior on Gaussian latent variables that generate the outputs. It is trained to maximize the conditional log-likelihood, and we formulate the variational learning objective of the CVAE in the framework of stochastic gradient variational Bayes (SGVB) [16]. In addition, we introduce several strategies, such as input noise-injection and multi-scale prediction training methods, to build a more robust prediction model.

In experiments, we demonstrate the effectiveness of our proposed algorithm in comparison to the deterministic neural network counterparts in generating diverse but realistic output predictions using stochastic inference. We demonstrate the importance of stochastic neurons in modeling the structured output when the input data is partially provided. Furthermore, we show that the proposed training schemes are complimentary, leading to strong pixel-level object segmentation and labeling performance on Caltech-UCSD Birds 200 and the subset of Labeled Faces in the Wild dataset.

In summary, the contribution of the paper is as follows:

- We propose CVAE and its variants that are trainable efficiently in the SGVB framework, and introduce novel strategies to enhance robustness of the models for structured prediction.
- We demonstrate the effectiveness of our proposed algorithm with Gaussian stochastic neurons in modeling multi-modal distribution of structured output variables.
- We achieve strong semantic object segmentation performance on CUB and LFW datasets.

The paper is organized as follows. We first review related work in Section 2. We provide preliminaries in Section 3 and develop our deep conditional generative model in Section 4. In Section 5, we evaluate our proposed models and report experimental results. Section 6 concludes the paper.

## 2    Related work

Since the recent success of supervised deep learning on large-scale visual recognition [17, 30, 27], there have been many approaches to tackle mid-level computer vision tasks, such as object detection [6, 26, 31, 9] and semantic segmentation [4, 3, 23, 22], using supervised deep learning techniques. Our work falls into this category of research in developing advanced algorithms for structured output prediction, but we incorporate the stochastic neurons to model the conditional distributions of complex output representation whose distribution possibly has multiple modes. In this sense, our work shares a similar motivation to the recent work on image segmentation tasks using hybrid models of CRF and Boltzmann machine [13, 21, 37]. Compared to these, our proposed model is an end-to-end system for segmentation using convolutional architecture and achieves significantly improved performance on challenging benchmark tasks.

Along with the recent breakthroughs in supervised deep learning methods, there has been a progress in deep generative models, such as deep belief networks [10, 20] and deep Boltzmann machines [25]. Recently, the advances in inference and learning algorithms for various deep generative models significantly enhanced this line of research [2, 7, 8, 18]. In particular, the variational learning framework of deep directed graphical model with Gaussian latent variables (e.g., variational auto-encoder [16, 15] and deep latent Gaussian models [24]) has been recently developed. Using the variational lower bound of the log-likelihood as the training objective and the reparameterization trick, these models can be easily trained via stochastic optimization. Our model builds upon this framework, but we focus on modeling the conditional distribution of output variables for structured prediction problems. Here, the main goal is not only to model the complex output representation but also to make a discriminative prediction. In addition, our model can effectively handle large-sized images by exploiting the convolutional architecture.

The stochastic feed-forward neural network (SFNN) [32] is a conditional directed graphical model with a combination of real-valued deterministic neurons and the binary stochastic neurons. The SFNN is trained using the Monte Carlo variant of generalized EM by drawing multiple samples from the feed-forward proposal distribution and weighing them differently with importance weights. Although our proposed Gaussian stochastic neural network (which will be described in Section 4.2) looks similar on surface, there are practical advantages in optimization of using Gaussian latent variables over the binary stochastic neurons. In addition, thanks to the recognition model used in our framework, it is sufficient to draw only a few samples during training, which is critical in training very deep convolutional networks.

## 3    Preliminary: Variational Auto-encoder

The variational auto-encoder (VAE) [16, 24] is a directed graphical model with certain types of latent variables, such as Gaussian latent variables. A generative process of the VAE is as follows: a set of latent variable $\mathbf{z}$ is generated from the prior distribution $p_\theta(\mathbf{z})$ and the data $\mathbf{x}$ is generated by the generative distribution $p_\theta(\mathbf{x}|\mathbf{z})$ conditioned on $\mathbf{z}$: $\mathbf{z} \sim p_\theta(\mathbf{z}), \mathbf{x} \sim p_\theta(\mathbf{x}|\mathbf{z})$.

In general, parameter estimation of directed graphical models is often challenging due to intractable posterior inference. However, the parameters of the VAE can be estimated efficiently in the stochastic gradient variational Bayes (SGVB) [16] framework, where the variational lower bound of the log-likelihood is used as a surrogate objective function. The variational lower bound is written as:

$$\log p_\theta(\mathbf{x}) = KL\left(q_\phi(\mathbf{z}|\mathbf{x})\|p_\theta(\mathbf{z}|\mathbf{x})\right) + \mathbb{E}_{q_\phi(\mathbf{z}|\mathbf{x})}\left[-\log q_\phi(\mathbf{z}|\mathbf{x}) + \log p_\theta(\mathbf{x}, \mathbf{z})\right] \quad (1)$$

$$\geq -KL\left(q_\phi(\mathbf{z}|\mathbf{x})\|p_\theta(\mathbf{z})\right) + \mathbb{E}_{q_\phi(\mathbf{z}|\mathbf{x})}\left[\log p_\theta(\mathbf{x}|\mathbf{z})\right] \quad (2)$$

In this framework, a proposal distribution $q_\phi(\mathbf{z}|\mathbf{x})$, which is also known as a "recognition" model, is introduced to approximate the true posterior $p_\theta(\mathbf{z}|\mathbf{x})$. The multilayer perceptrons (MLPs) are used to model the recognition and the generation models. Assuming Gaussian latent variables, the first term of Equation (2) can be marginalized, while the second term is not. Instead, the second term can be approximated by drawing samples $\mathbf{z}^{(l)}$ ($l = 1, ..., L$) by the recognition distribution $q_\phi(\mathbf{z}|\mathbf{x})$, and the empirical objective of the VAE with Gaussian latent variables is written as follows:

$$\widetilde{\mathcal{L}}_{\text{VAE}}(\mathbf{x}; \theta, \phi) = -KL\left(q_\phi(\mathbf{z}|\mathbf{x})\|p_\theta(\mathbf{z})\right) + \frac{1}{L}\sum_{l=1}^{L}\log p_\theta(\mathbf{x}|\mathbf{z}^{(l)}), \tag{3}$$

where $\mathbf{z}^{(l)} = g_\phi(\mathbf{x}, \epsilon^{(l)})$, $\epsilon^{(l)} \sim \mathcal{N}(\mathbf{0}, \mathbf{I})$. Note that the recognition distribution $q_\phi(\mathbf{z}|\mathbf{x})$ is reparameterized with a deterministic, differentiable function $g_\phi(\cdot, \cdot)$, whose arguments are data $\mathbf{x}$ and the noise variable $\epsilon$. This trick allows error backpropagation through the Gaussian latent variables, which is essential in VAE training as it is composed of multiple MLPs for recognition and generation models. As a result, the VAE can be trained efficiently using stochastic gradient descent (SGD).

## 4 Deep Conditional Generative Models for Structured Output Prediction

As illustrated in Figure 1, there are three types of variables in a deep conditional generative model (CGM): input variables $\mathbf{x}$, output variables $\mathbf{y}$, and latent variables $\mathbf{z}$. The conditional generative process of the model is given in Figure 1(b) as follows: for given observation $\mathbf{x}$, $\mathbf{z}$ is drawn from the prior distribution $p_\theta(\mathbf{z}|\mathbf{x})$, and the output $\mathbf{y}$ is generated from the distribution $p_\theta(\mathbf{y}|\mathbf{x}, \mathbf{z})$. Compared to the baseline CNN (Figure 1(a)), the latent variables $\mathbf{z}$ allow for modeling multiple modes in conditional distribution of output variables $\mathbf{y}$ given input $\mathbf{x}$, making the proposed CGM suitable for modeling one-to-many mapping. The prior of the latent variables $\mathbf{z}$ is modulated by the input $\mathbf{x}$ in our formulation; however, the constraint can be easily relaxed to make the latent variables statistically independent of input variables, i.e., $p_\theta(\mathbf{z}|\mathbf{x}) = p_\theta(\mathbf{z})$ [15].

Deep CGMs are trained to maximize the conditional log-likelihood. Often the objective function is intractable, and we apply the SGVB framework to train the model. The variational lower bound of the model is written as follows (complete derivation can be found in the supplementary material):

$$\log p_\theta(\mathbf{y}|\mathbf{x}) \geq -KL\left(q_\phi(\mathbf{z}|\mathbf{x}, \mathbf{y})\|p_\theta(\mathbf{z}|\mathbf{x})\right) + \mathbb{E}_{q_\phi(\mathbf{z}|\mathbf{x}, \mathbf{y})}\left[\log p_\theta(\mathbf{y}|\mathbf{x}, \mathbf{z})\right] \tag{4}$$

and the empirical lower bound is written as:

$$\widetilde{\mathcal{L}}_{\text{CVAE}}(\mathbf{x}, \mathbf{y}; \theta, \phi) = -KL\left(q_\phi(\mathbf{z}|\mathbf{x}, \mathbf{y})\|p_\theta(\mathbf{z}|\mathbf{x})\right) + \frac{1}{L}\sum_{l=1}^{L}\log p_\theta(\mathbf{y}|\mathbf{x}, \mathbf{z}^{(l)}), \tag{5}$$

where $\mathbf{z}^{(l)} = g_\phi(\mathbf{x}, \mathbf{y}, \epsilon^{(l)})$, $\epsilon^{(l)} \sim \mathcal{N}(\mathbf{0}, \mathbf{I})$ and $L$ is the number of samples. We call this model *conditional variational auto-encoder*[1] (CVAE). The CVAE is composed of multiple MLPs, such as recognition network $q_\phi(\mathbf{z}|\mathbf{x}, \mathbf{y})$, (conditional) prior network $p_\theta(\mathbf{z}|\mathbf{x})$, and generation network $p_\theta(\mathbf{y}|\mathbf{x}, \mathbf{z})$. In designing the network architecture, we build the network components of the CVAE on top of the baseline CNN. Specifically, as shown in Figure 1(d), not only the direct input $\mathbf{x}$, but also the initial guess $\hat{\mathbf{y}}$ made by the CNN are fed into the prior network. Such a recurrent connection has been applied for structured output prediction problems [23, 13, 28] to sequentially update the prediction by revising the previous guess while effectively deepening the convolutional network. We also found that a recurrent connection, even one iteration, showed significant performance improvement. Details about network architectures can be found in the supplementary material.

### 4.1 Output inference and estimation of the conditional likelihood

Once the model parameters are learned, we can make a prediction of an output $\mathbf{y}$ from an input $\mathbf{x}$ by following the generative process of the CGM. To evaluate the model on structured output prediction tasks (i.e., in testing time), we can measure a prediction accuracy by performing a deterministic inference without sampling $\mathbf{z}$, i.e., $\mathbf{y}^* = \arg\max_{\mathbf{y}} p_\theta(\mathbf{y}|\mathbf{x}, \mathbf{z}^*)$, $\mathbf{z}^* = \mathbb{E}\left[\mathbf{z}|\mathbf{x}\right]$.[2]

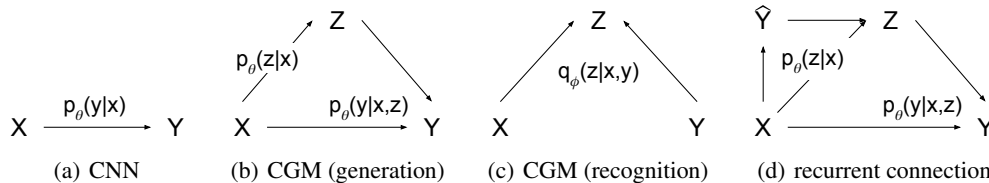

<center>(a) CNN     (b) CGM (generation)     (c) CGM (recognition)     (d) recurrent connection</center>

Figure 1: Illustration of the conditional graphical models (CGMs). (a) the predictive process of output $Y$ for the baseline CNN; (b) the generative process of CGMs; (c) an approximate inference of $Z$ (also known as recognition process [16]); (d) the generative process with recurrent connection.

Another way to evaluate the CGMs is to compare the conditional likelihoods of the test data. A straightforward approach is to draw samples $\mathbf{z}$'s using the prior network and take the average of the likelihoods. We call this method the Monte Carlo (MC) sampling:

$$p_\theta(\mathbf{y}|\mathbf{x}) \approx \frac{1}{S} \sum_{s=1}^{S} p_\theta(\mathbf{y}|\mathbf{x}, \mathbf{z}^{(s)}), \ \ \mathbf{z}^{(s)} \sim p_\theta(\mathbf{z}|\mathbf{x}) \tag{6}$$

It usually requires a large number of samples for the Monte Carlo log-likelihood estimation to be accurate. Alternatively, we use the importance sampling to estimate the conditional likelihoods [24]:

$$p_\theta(\mathbf{y}|\mathbf{x}) \approx \frac{1}{S} \sum_{s=1}^{S} \frac{p_\theta(\mathbf{y}|\mathbf{x}, \mathbf{z}^{(s)}) p_\theta(\mathbf{z}^{(s)}|\mathbf{x})}{q_\phi(\mathbf{z}^{(s)}|\mathbf{x}, \mathbf{y})}, \ \ \mathbf{z}^{(s)} \sim q_\phi(\mathbf{z}|\mathbf{x}, \mathbf{y}) \tag{7}$$

### 4.2 Learning to predict structured output

Although the SGVB learning framework has shown to be effective in training deep generative models [16, 24], the conditional *auto-encoding* of output variables at training may not be optimal to make a prediction at testing in deep CGMs. In other words, the CVAE uses the recognition network $q_\phi(\mathbf{z}|\mathbf{x}, \mathbf{y})$ at training, but it uses the prior network $p_\theta(\mathbf{z}|\mathbf{x})$ at testing to draw samples $\mathbf{z}$'s and make an output prediction. Since $\mathbf{y}$ is given as an input for the recognition network, the objective at training can be viewed as a *reconstruction* of $\mathbf{y}$, which is an easier task than prediction. The negative KL divergence term in Equation (5) tries to close the gap between two pipelines, and one could consider allocating more weights on the negative KL term of an objective function to mitigate the discrepancy in encoding of latent variables at training and testing, i.e., $-(1 + \beta)KL\left(q_\phi(\mathbf{z}|\mathbf{x}, \mathbf{y})\|p_\theta(\mathbf{z}|\mathbf{x})\right)$ with $\beta \geq 0$. However, we found this approach ineffective in our experiments.

Instead, we propose to train the networks in a way that the prediction pipelines at training and testing are consistent. This can be done by setting the recognition network the same as the prior network, i.e., $q_\phi(\mathbf{z}|\mathbf{x}, \mathbf{y}) = p_\theta(\mathbf{z}|\mathbf{x})$, and we get the following objective function:

$$\widetilde{\mathcal{L}}_{\text{GSNN}}(\mathbf{x}, \mathbf{y}; \theta, \phi) = \frac{1}{L} \sum_{l=1}^{L} \log p_\theta(\mathbf{y}|\mathbf{x}, \mathbf{z}^{(l)}), \text{ where } \mathbf{z}^{(l)} = g_\theta(\mathbf{x}, \epsilon^{(l)}), \ \epsilon^{(l)} \sim \mathcal{N}(\mathbf{0}, \mathbf{I}) \tag{8}$$

We call this model *Gaussian stochastic neural network* (GSNN).[3] Note that the GSNN can be derived from the CVAE by setting the recognition network and the prior network equal. Therefore, the learning tricks, such as reparameterization trick, of the CVAE can be used to train the GSNN. Similarly, the inference (at testing) and the conditional likelihood estimation are the same as those of CVAE. Finally, we combine the objective functions of two models to obtain a hybrid objective:

$$\widetilde{\mathcal{L}}_{\text{hybrid}} = \alpha \widetilde{\mathcal{L}}_{\text{CVAE}} + (1 - \alpha) \widetilde{\mathcal{L}}_{\text{GSNN}}, \tag{9}$$

where $\alpha$ balances the two objectives. Note that when $\alpha = 1$, we recover the CVAE objective; when $\alpha = 0$, the trained model will be simply a GSNN without the recognition network.

### 4.3 CVAE for image segmentation and labeling

Semantic segmentation [5, 23, 6] is an important structured output prediction task. In this section, we provide strategies to train a robust prediction model for semantic segmentation problems. Specifically, to learn a high-capacity neural network that can be generalized well to unseen data, we propose to train the network with 1) multi-scale prediction objective and 2) structured input noise.

### 4.3.1 Training with multi-scale prediction objective

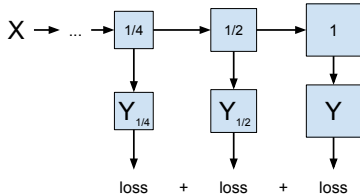

Figure 2: Multi-scale prediction.

As the image size gets larger (e.g., $128 \times 128$), it becomes more challenging to make a fine-grained pixel-level prediction (e.g., image reconstruction, semantic label prediction). The multi-scale approaches have been used in the sense of forming a multi-scale image pyramid for an input [5], but not much for multi-scale output prediction. Here, we propose to train the network to predict outputs at different scales. By doing so, we can make a global-to-local, coarse-to-fine-grained prediction of pixel-level semantic labels. Figure 2 describes the multi-scale prediction at 3 different scales ($^1/_4$, $^1/_2$, and original) for the training.

### 4.3.2 Training with input omission noise

Adding noise to neurons is a widely used technique to regularize deep neural networks during the training [17, 29]. Similarly, we propose a simple regularization technique for semantic segmentation: corrupt the input data $\mathbf{x}$ into $\tilde{\mathbf{x}}$ according to noise process and optimize the network with the following objective: $\widetilde{\mathcal{L}}(\tilde{\mathbf{x}}, \mathbf{y})$. The noise process could be arbitrary, but for semantic image segmentation, we consider random block omission noise. Specifically, we randomly generate a squared mask of width and height less than $40\%$ of the image width and height, respectively, at random position and set pixel values of the input image inside the mask to $0$. This can be viewed as providing more challenging output prediction task during training that simulates block occlusion or missing input. The proposed training strategy also is related to the denoising training methods [34], but in our case, we inject noise to the input data only and do not reconstruct the missing input.

## 5 Experiments

We demonstrate the effectiveness of our approach in modeling the distribution of the structured output variables. For the proof of concept, we create an artificial experimental setting for structured output prediction using MNIST database [19]. Then, we evaluate the proposed CVAE models on several benchmark datasets for visual object segmentation and labeling, such as Caltech-UCSD Birds (CUB) [36] and Labeled Faces in the Wild (LFW) [12]. Our implementation is based on MatConvNet [33], a MATLAB toolbox for convolutional neural networks, and Adam [14] for adaptive learning rate scheduling algorithm of SGD optimization.

### 5.1 Toy example: MNIST

To highlight the importance of probabilistic inference through stochastic neurons for structured output variables, we perform an experiment using MNIST database. Specifically, we divide each digit image into four quadrants, and take one, two, or three quadrant(s) as an input and the remaining quadrants as an output.[4] As we increase the number of quadrants for an output, the input to output mapping becomes more diverse (in terms of one-to-many mapping).

We trained the proposed models (CVAE, GSNN) and the baseline deep neural network and compare their performance. The same network architecture, the MLP with two-layers of $1,000$ ReLUs for recognition, conditional prior, or generation networks, followed by $200$ Gaussian latent variables, was used for all the models in various experimental settings. The early stopping is used during the training based on the estimation of the conditional likelihoods on the validation set.

| negative CLL | 1 quadrant | | 2 quadrants | | 3 quadrants | |
|---|---|---|---|---|---|---|
| | validation | test | validation | test | validation | test |
| NN (baseline) | 100.03 | 99.75 | 62.14 | 62.18 | 26.01 | 25.99 |
| GSNN (Monte Carlo) | 100.03 | 99.82 | 62.48 | 62.41 | 26.20 | 26.29 |
| CVAE (Monte Carlo) | 68.62 | 68.39 | 45.57 | 45.34 | 20.97 | 20.96 |
| CVAE (Importance Sampling) | **64.05** | **63.91** | **44.96** | **44.73** | **20.97** | **20.95** |
| Performance gap | 35.98 | 35.91 | 17.51 | 17.68 | 5.23 | 5.33 |
| - per pixel | 0.061 | 0.061 | 0.045 | 0.045 | 0.027 | 0.027 |

Table 1: The negative CLL on MNIST database. We increase the number of quadrants for an input from 1 to 3. The performance gap between CVAE (importance sampling) and NN is reported.

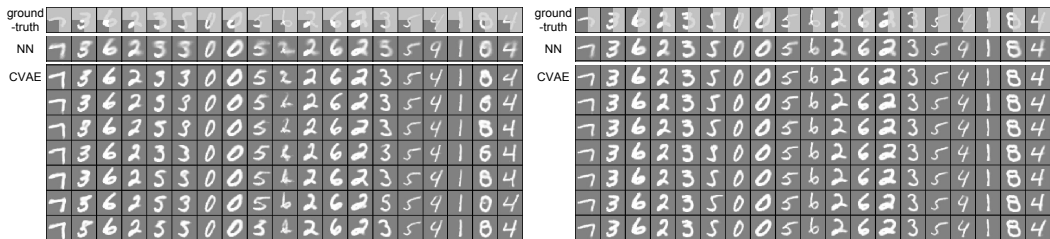

Figure 3: Visualization of generated samples with (left) 1 quadrant and (right) 2 quadrants for an input. We show in each row the input and the ground truth output overlaid with gray color (first), samples generated by the baseline NNs (second), and samples drawn from the CVAEs (rest).

For qualitative analysis, we visualize the generated output samples in Figure 3. As we can see, the baseline NNs can only make a single deterministic prediction, and as a result the output looks blurry and doesn't look realistic in many cases. In contrast, the samples generated by the CVAE models are more realistic and diverse in shape; sometimes they can even change their identity (digit labels), such as from 3 to 5 or from 4 to 9, and vice versa.

We also provide a quantitative evidence by estimating the conditional log-likelihoods (CLLs) in Table 1. The CLLs of the proposed models are estimated in two ways as described in Section 4.1. For the MC estimation, we draw $10,000$ samples per example to get an accurate estimate. For the importance sampling, however, $100$ samples per example were enough to obtain an accurate estimation of the CLL. We observed that the estimated CLLs of the CVAE significantly outperforms the baseline NN. Moreover, as measured by the per pixel performance gap, the performance improvement becomes more significant as we use smaller number of quadrants for an input, which is expected as the input-output mapping becomes more diverse.

## 5.2 Visual Object Segmentation and Labeling

**Caltech-UCSD Birds (CUB)** database [36] includes $6,033$ images of birds from 200 species with annotations such as a bounding box of birds and a segmentation mask. Later, Yang et al. [37] annotated these images with more fine-grained segmentation masks by cropping the bird patches using the bounding boxes and resized them into $128 \times 128$ pixels. The training/test split proposed in [36] was used in our experiment, and for validation purpose, we partition the training set into 10 folds and cross-validated with the mean intersection over union (IoU) score over the folds. The final prediction on the test set was made by averaging the posterior from ensemble of 10 networks that are trained on each of the 10 folds separately. We increase the number of training examples via "data augmentation" by horizontally flipping the input and output images.

We extensively evaluate the variations of our proposed methods, such as CVAE, GSNN, and the hybrid model, and provide a summary results on segmentation mask prediction task in Table 2. Specifically, we report the performance of the models with different network architectures and training methods (e.g., multi-scale prediction or noise-injection training).

First, we note that the baseline CNN already beat the previous state-of-the-art that is obtained by the max-margin Boltzmann machine (MMBM; pixel accuracy: 90.42, IoU: 75.92 with GraphCut for post-processing) [37] even without post-processing. On top of that, we observed significant performance improvement with our proposed deep CGMs.[5] In terms of prediction accuracy, the GSNN performed the best among our proposed models, and performed even better when it is trained with hybrid objective function. In addition, the noise-injection training (Section 4.3) further improves the performance. Compared to the baseline CNN, the proposed deep CGMs significantly reduce the prediction error, e.g., $21\%$ reduction in test pixel-level accuracy at the expense of $60\%$ more time for inference.[6] Finally, the performance of our two winning entries (GSNN and hybrid) on the validation sets are both significantly better than their deterministic counterparts (GDNN) with p-values less than $0.05$, which suggests the benefit of stochastic latent variables.

| Model (training) | CUB (val) | | CUB (test) | | LFW | |
|---|---|---|---|---|---|---|
| | pixel | IoU | pixel | IoU | pixel (val) | pixel (test) |
| MMBM [37] | – | – | 90.42 | 75.92 | – | – |
| GLOC [13] | – | – | – | – | – | 90.70 |
| CNN (baseline) | 91.17 $\pm0.09$ | 79.64 $\pm0.24$ | 92.30 | 81.90 | 92.09 $\pm0.13$ | 91.90 $\pm0.08$ |
| CNN (msc) | 91.37 $\pm0.09$ | 80.09 $\pm0.25$ | 92.52 | 82.43 | 92.19 $\pm0.10$ | 92.05 $\pm0.06$ |
| GDNN (msc) | 92.25 $\pm0.09$ | 81.89 $\pm0.21$ | 93.24 | 83.96 | 92.72 $\pm0.12$ | 92.54 $\pm0.04$ |
| GSNN (msc) | 92.46 $\pm0.07$ | 82.31 $\pm0.19$ | 93.39 | 84.26 | 92.88 $\pm0.08$ | 92.61 $\pm0.09$ |
| CVAE (msc) | 92.24 $\pm0.09$ | 81.86 $\pm0.23$ | 93.03 | 83.53 | 92.80 $\pm0.30$ | 92.62 $\pm0.06$ |
| hybrid (msc) | 92.60 $\pm0.08$ | 82.57 $\pm0.26$ | 93.35 | 84.16 | 92.95 $\pm0.21$ | 92.77 $\pm0.06$ |
| GDNN (msc, NI) | 92.92 $\pm0.07$ | 83.20 $\pm0.19$ | 93.78 | 85.07 | **93.59** $\pm0.12$ | **93.25** $\pm0.06$ |
| GSNN (msc, NI) | **93.09** $\pm0.09$ | **83.62** $\pm0.21$ | **93.91** | **85.39** | **93.71** $\pm0.09$ | **93.51** $\pm0.07$ |
| CVAE (msc, NI) | 92.72 $\pm0.08$ | 82.90 $\pm0.22$ | 93.48 | 84.47 | 93.29 $\pm0.17$ | 93.22 $\pm0.08$ |
| hybrid (msc, NI) | **93.05** $\pm0.07$ | **83.49** $\pm0.19$ | **93.78** | **85.07** | **93.69** $\pm0.12$ | **93.42** $\pm0.07$ |

Table 2: Mean and standard error of labeling accuracy on CUB and LFW database. The performance of the best or statistically similar (i.e., p-value $\geq$ 0.05 to the best performing model) models are bold-faced. "msc" refers multi-scale prediction training and "NI" refers the noise-injection training.

| Models | CUB (val) | CUB (test) | LFW (val) | LFW (test) |
|---|---|---|---|---|
| CNN (baseline) | 4269.43 $\pm130.90$ | 4329.94 $\pm91.71$ | 6370.63 $\pm790.53$ | 6434.09 $\pm756.57$ |
| GDNN (msc, NI) | 3386.19 $\pm44.11$ | 3450.41 $\pm33.36$ | 4710.46 $\pm192.77$ | 5170.26 $\pm166.81$ |
| GSNN (msc, NI) | 3400.24 $\pm59.42$ | 3461.87 $\pm25.57$ | 4582.96 $\pm225.62$ | 4829.45 $\pm96.98$ |
| CVAE (msc, NI) | **801.48** $\pm4.34$ | **801.31** $\pm1.86$ | **1262.98** $\pm64.43$ | **1267.58** $\pm57.92$ |
| hybrid (msc, NI) | 1019.93 $\pm8.46$ | 1021.44 $\pm4.81$ | 1836.98 $\pm127.53$ | 1867.47 $\pm111.26$ |

Table 3: Mean and standard error of negative CLL on CUB and LFW database. The performance of the best and statistically similar models are bold-faced.

We also evaluate the negative CLL and summarize the results in Table 3. As expected, the proposed CGMs significantly outperform the baseline CNN while the CVAE showed the highest CLL.

**Labeled Faces in the Wild (LFW)** database [12] has been widely used for face recognition and verification benchmark. As mentioned in [11], the face images that are segmented and labeled into semantically meaningful region labels (e.g., hair, skin, clothes) can greatly help understanding of the image through the visual attributes, which can be easily obtained from the face shape.

Following region labeling protocols [35, 13], we evaluate the performance of face parts labeling on the subset of LFW database [35], which contains $1,046$ images that are labeled into 4 semantic categories, such as hair, skin, clothes, and background. We resized images into $128 \times 128$ and used the same network architecture to the one used in the CUB experiment.

We provide summary results of pixel-level segmentation accuracy in Table 2 and the negative CLL in Table 3. We observe a similar trend as previously shown for the CUB database; the proposed deep CGMs outperform the baseline CNN in terms of segmentation accuracy as well as CLL. However, although the accuracies of the CGM variants are higher, the performance of GDNN was not significantly behind than those of GSNN and hybrid models. This may be because the level of variations in the output space of LFW database is less than that of CUB database as the face shapes are more similar and better aligned across examples. Finally, our methods significantly outperform other existing methods, which report 90.0% in [35] or 90.7% in [13], setting the state-of-the-art performance on the LFW segmentation benchmark.

## 5.3 Object Segmentation with Partial Observations

We experimented on object segmentation under uncertainties (e.g., partial input and output observations) to highlight the importance of recognition network in CVAE and the stochastic neurons for missing value imputation. We randomly omit the input pixels at different levels of omission noise (25%, 50%, 70%) and different block sizes (1, 4, 8), and the task is to predict the output segmentation labels for the omitted pixel locations while given the partial labels for the observed input pixels. This can also be viewed as a segmentation task with noisy or partial observations (e.g., occlusions).

To make a prediction for CVAE with partial output observation ($\mathbf{y}_o$), we perform iterative inference of unobserved output ($\mathbf{y}_u$) and the latent variables ($\mathbf{z}$) (in a similar fashion to [24]), i.e.,

$$\mathbf{y}_u \sim p_\theta(\mathbf{y}_u|\mathbf{x}, \mathbf{z}) \leftrightarrow \mathbf{z} \sim q_\phi(\mathbf{z}|\mathbf{x}, \mathbf{y}_o, \mathbf{y}_u). \tag{10}$$

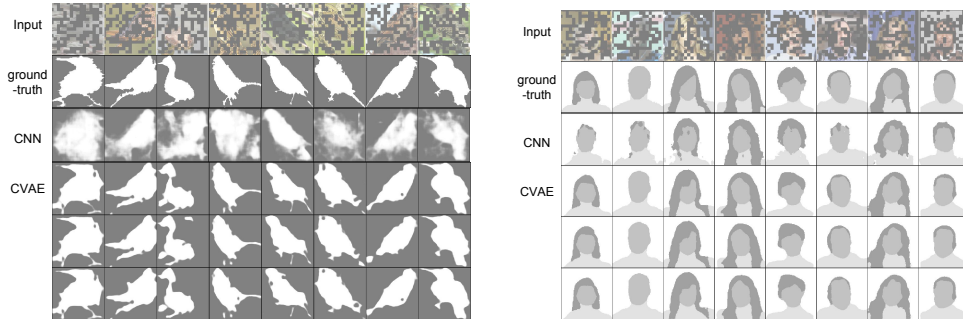

Figure 4: Visualization of the conditionally generated samples: (first row) input image with omission noise (noise level: 50%, block size: 8), (second row) ground truth segmentation, (third) prediction by GDNN, and (fourth to sixth) the generated samples by CVAE on CUB (left) and LFW (right).

We report the summary results in Table 4. The CVAE performs well even when the noise level is high (e.g., 50%), where the GDNN significantly fails. This is because the CVAE utilizes the partial segmentation information to iteratively refine the prediction of the rest. We visualize the generated samples at noise level of 50% in Figure 4. The prediction made by the GDNN is blurry, but the samples generated by the CVAE are sharper while maintaining reasonable shapes. This suggests that the CVAE can also be potentially useful for interactive segmentation (i.e., by iteratively incorporating partial output labels).

| Dataset | | CUB (IoU) | | LFW (pixel) | |
|---|---|---|---|---|---|
| noise level | block size | GDNN | CVAE | GDNN | CVAE |
| 25% | 1 | 89.37 | **98.52** | 96.93 | **99.22** |
| | 4 | 88.74 | **98.07** | 96.55 | **99.09** |
| | 8 | 90.72 | **96.78** | 97.14 | **98.73** |
| 50% | 1 | 74.95 | **95.95** | 91.84 | **97.29** |
| | 4 | 70.48 | **94.25** | 90.87 | **97.08** |
| | 8 | 76.07 | **89.10** | 92.68 | **96.15** |
| 70% | 1 | 62.11 | **89.44** | 85.27 | **89.71** |
| | 4 | 57.68 | **84.36** | 85.70 | **93.16** |
| | 8 | 63.59 | **76.87** | 87.83 | **92.06** |

Table 4: Segmentation results with omission noise on CUB and LFW database. We report the pixel-level accuracy on the first validation set.

# 6 Conclusion

Modeling multi-modal distribution of the structured output variables is an important research question to achieve good performance on structured output prediction problems. In this work, we proposed stochastic neural networks for structured output prediction based on the conditional deep generative model with Gaussian latent variables. The proposed model is scalable and efficient in inference and learning. We demonstrated the importance of probabilistic inference when the distribution of output space has multiple modes, and showed strong performance in terms of segmentation accuracy, estimation of conditional log-likelihood, and visualization of generated samples.

**Acknowledgments**   This work was supported in part by ONR grant N00014-13-1-0762 and NSF CAREER grant IIS-1453651. We thank NVIDIA for donating a Tesla K40 GPU.

## Footnotes

[1]Although the model is not trained to reconstruct the input $\mathbf{x}$, our model can be viewed as a type of VAE that performs auto-encoding of the output variables $\mathbf{y}$ conditioned on the input $\mathbf{x}$ at training time.

[2]Alternatively, we can draw multiple $\mathbf{z}$'s from the prior distribution and use the average of the posteriors to make a prediction, i.e., $\mathbf{y}^* = \arg\max_{\mathbf{y}} \frac{1}{L}\sum_{l=1}^{L} p_\theta(\mathbf{y}|\mathbf{x}, \mathbf{z}^{(l)})$, $\mathbf{z}^{(l)} \sim p_\theta(\mathbf{z}|\mathbf{x})$.

[3]If we assume a covariance matrix of auxiliary Gaussian latent variables $\epsilon$ to $\mathbf{0}$, we have a deterministic counterpart of GSNN, which we call a Gaussian deterministic neural network (GDNN).

[4]Similar experimental setting has been used in the multimodal learning framework, where the left- and right halves of the digit images are used as two data modalities [1, 28].

[5]As in the case of baseline CNNs, we found that using the multi-scale prediction was consistently better than the single-scale counterpart for all our models. So, we used the multi-scale prediction by default.

[6]Mean inference time per image: $2.32$ (ms) for CNN and $3.69$ (ms) for deep CGMs, measured using GeForce GTX TITAN X card with MatConvNet; we provide more information in the supplementary material.

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
