[Supplementary Material]

# Supplementary Material:
# Learning Structured Output Representation using Deep Conditional Generative Models

**Kihyuk Sohn**$^{*\dagger}$  **Xinchen Yan**$^{\dagger}$  **Honglak Lee**$^{\dagger}$
$^{*}$ NEC Laboratories America, Inc.
$^{\dagger}$ University of Michigan, Ann Arbor
ksohn@nec-labs.com, {xcyan,honglak}@umich.edu

## S1  Variational Lower Bound of Conditional Log-Likelihood

We provide a derivation for variational lower bound of the conditional log-likelihood (Equation (4) in the main paper):

$$\log p_\theta(\mathbf{y}|\mathbf{x}) = KL\left(q_\phi(\mathbf{z}|\mathbf{x},\mathbf{y}) \| p_\theta(\mathbf{z}|\mathbf{x},\mathbf{y})\right) + \mathbb{E}_{q_\phi(\mathbf{z}|\mathbf{x},\mathbf{y})}\left[-\log q_\phi(\mathbf{z}|\mathbf{x},\mathbf{y}) + \log p_\theta(\mathbf{y},\mathbf{z}|\mathbf{x})\right] \quad \text{(S1)}$$

$$\geq \mathbb{E}_{q_\phi(\mathbf{z}|\mathbf{x},\mathbf{y})}\left[-\log q_\phi(\mathbf{z}|\mathbf{x},\mathbf{y}) + \log p_\theta(\mathbf{y},\mathbf{z}|\mathbf{x})\right] \quad \text{(S2)}$$

$$= \mathbb{E}_{q_\phi(\mathbf{z}|\mathbf{x},\mathbf{y})}\left[-\log q_\phi(\mathbf{z}|\mathbf{x},\mathbf{y}) + \log p_\theta(\mathbf{z}|\mathbf{x})\right] + \mathbb{E}_{q_\phi(\mathbf{z}|\mathbf{x},\mathbf{y})}\left[\log p_\theta(\mathbf{y}|\mathbf{x},\mathbf{z})\right] \quad \text{(S3)}$$

$$= -KL\left(q_\phi(\mathbf{z}|\mathbf{x},\mathbf{y}) \| p_\theta(\mathbf{z}|\mathbf{x})\right) + \mathbb{E}_{q_\phi(\mathbf{z}|\mathbf{x},\mathbf{y})}\left[\log p_\theta(\mathbf{y}|\mathbf{x},\mathbf{z})\right] \quad \text{(S4)}$$

## S2  Details of Network Architecture

We provide more detailed information of our network architecture used for experiments on LFW (4-way segmentation task) database in Tables S1, S2, and S3. The model architecture used for experiments on CUB database is almost the same except that the task is 2-way segmentation.

**Connecting CNN and prior network.** We provide both input data $\mathbf{x}$ and the output of the CNN as an input for prior network. Specifically, we concatenate the data $\mathbf{x}$ of size $128 \times 128 \times 3$ and the CNN output (i.e., CNN followed by softmax classifier) of size $128 \times 128 \times 4$ along the channels, and feed concatenated data of size $128 \times 128 \times 7$ as an input of the prior network (Figure S1).

**Final output prediction.** As shown in Figure S1, the final output prediction is made by element-wise summing the output of two convolutional networks, which are the CNN and the generation network, followed by softmax classifier. We note that the convolution filters in two yellow boxes are shared, so that the initial guess made by CNN can still be reasonable.

**Analysis on computation time.** We perform an analysis on computation time of our proposed method and compare with the baseline CNN model. We measure the mean processing times for the forward pass and the backward pass. Note that the forward pass in testing time is the same across all the proposed deep CGMs since recognition network is only used during the training. We summarize the mean processing times per single image from LFW database in Table S4. In all our experiments, we used mini-batch size of 32 and the time is averaged over 100 iterations. All deep CGMs are trained with multi-scale prediction method, but the output prediction is made only for the highest resolution at test time.

| layer | op. | size-in | size-out | kernel |
|---|---|---|---|---|
| 1 | conv | 128×128×3 | 64×64×64 | 9×9×3 |
| | pool | 64×64×64 | 32×32×64 | 2×2×1 |
| | relu | 32×32×64 | 32×32×64 | – |
| 2 | conv | 32×32×64 | 32×32×96 | 5×5×64 |
| | pool | 32×32×96 | 16×16×96 | 2×2×1 |
| | relu | 16×16×96 | 16×16×96 | – |
| 3 | conv | 16×16×96 | 16×16×128 | 3×3×96 |
| | relu | 16×16×128 | 16×16×128 | – |
| 4 | conv | 16×16×128 | 16×16×128 | 3×3×128 |
| | relu | 16×16×128 | 16×16×128 | – |
| 5 | conv | 16×16×128 | 16×16×96 | 3×3×96 |
| | unpool | 16×16×96 | 32×32×96 | 2×2×1 |
| | relu | 32×32×96 | 32×32×96 | – |
| 6 | conv | 32×32×96 | 32×32×64 | 5×5×64 |
| | unpool | 32×32×64 | 64×64×64 | 2×2×1 |
| | relu | 64×64×64 | 64×64×64 | – |
| 7 | conv | 64×64×64 | 64×64×48 | 5×5×48 |
| | unpool | 64×64×48 | 128×128×48 | 2×2×1 |
| | relu | 128×128×48 | 128×128×48 | – |
| 8 | conv | 128×128×48 | 128×128×4 | 9×9×48 |
| | softmax | 128×128×4 | 128×128×4 | – |

Table S1: CNN definition. "conv" refers the layer including convolution followed by ReLU.

| layer | op. | size-in | size-out | kernel |
|---|---|---|---|---|
| 1 | conv | 128×128×7 | 64×64×64 | 9×9×7 |
| | pool | 64×64×64 | 32×32×64 | 2×2×1 |
| | relu | 32×32×64 | 32×32×64 | – |
| 2 | conv | 32×32×64 | 32×32×96 | 5×5×64 |
| | pool | 32×32×96 | 16×16×96 | 2×2×1 |
| | relu | 16×16×96 | 16×16×96 | – |
| 3 | conv | 16×16×96 | 16×16×128 | 3×3×96 |
| | pool | 16×16×128 | 8×8×128 | 2×2×1 |
| | relu | 8×8×128 | 8×8×128 | – |
| 4 | conv$^g$ | 8×8×128 | 8×8×32 | 3×3×128 |
| | sampling | 8×8×32 | 8×8×32 | – |

Table S2: Prior and recognition networks definition. "conv$^g$" refers the layer that outputs Gaussian latent variables, and it includes convolution for mean and standard deviation units followed by gaussian sampling ("sampling").

| layer | op. | size-in | size-out | kernel |
|---|---|---|---|---|
| 1 | conv | 8×8×32 | 8×8×96 | 3×3×32 |
| | unpool | 8×8×96 | 16×16×96 | 2×2×1 |
| | relu | 16×16×96 | 16×16×96 | – |
| 2 | conv | 16×16×96 | 16×16×64 | 5×5×96 |
| | unpool | 16×16×64 | 32×32×64 | 2×2×1 |
| | relu | 32×32×64 | 32×32×64 | – |
| 3 | conv | 32×32×64 | 32×32×48 | 5×5×64 |
| | unpool | 32×32×48 | 64×64×48 | 2×2×1 |
| | relu | 64×64×48 | 64×64×48 | – |
| 4 | conv | 64×64×48 | 64×64×48 | 5×5×48 |
| | unpool | 64×64×48 | 128×128×48 | 2×2×1 |
| | relu | 128×128×48 | 128×128×48 | – |
| 5 | conv | 128×128×48 | 128×128×4 | 9×9×48 |
| | softmax | 128×128×4 | 128×128×4 | – |

Table S3: Generation network definition.

Image (X) + Label (Y) 128x128x7 — Conv (9x9) Pool (2x2) 32x32x64 — Conv (5x5) Pool (2x2) 16x16x96 — Conv (3x3) Pool (2x2) 8x8x128 — $\text{Conv}^g$ (3x3) Sampling (Z) 8x8x128

Recognition network

Generation network

Conv (3x3) Unpool (2x2) 16x16x96 — Conv (5x5) Unpool (2x2) 32x32x64 — Conv (5x5) Unpool (2x2) 64x64x48 — Conv (5x5) Unpool (2x2) 128x128x48

Prior network

Image (X) + CNN out (Ŷ) 128x128x7 — Conv (9x9) Pool (2x2) 32x32x64 — Conv (5x5) Pool (2x2) 16x16x96 — Conv (3x3) Pool (2x2) 8x8x128 — $\text{Conv}^g$ (3x3) Sampling (Z) 8x8x128

$\text{Conv}^2$ (9x9) Softmax (Ŷ) 128x128x4

Image (X) 128x128x3 — Conv (9x9) Pool (2x2) 32x32x64 — Conv (5x5) Pool (2x2) 16x16x96 — Conv (3x3) 16x16x128 — Conv (3x3) 16x16x128 — Conv (3x3) Unpool (2x2) 32x32x96 — Conv (5x5) Unpool (2x2) 64x64x64 — Conv (5x5) Unpool (2x2) 128x128x48 — $\text{Conv}^1$ (9x9) Softmax (Y) 128x128x4

CNN

Figure S1: Model architecture of our deep CGM for 4-way semantic segmentation problem. ReLU is followed by convolution layers except $\text{conv}^g$ layers. For baseline CNN model, only the output of CNN is used for prediction. The convolutional filter weights for classifiers (yellow box) are shared.

| Phase | CNN | GDNN | GSNN | CVAE | hybrid |
|---|---|---|---|---|---|
| Forward (test) | 2.32 | 3.69 | 3.69 | 3.69 | 3.69 |
| Forward (train) | 2.32 | 4.22 | 4.80 | 5.48 | 6.55 |
| Backward (train) | 10.12 | 16.58 | 16.87 | 27.32 | 31.49 |

Table S4: Comparison of average processing time per single image among different network architectures and learning algorithms. All deep CGMs are trained with multi-scale prediction. We measure the computation time using GeForce GTX TITAN X card using mini-batch size of 32 and the time is averaged over 100 iterations. Times are in millisecond (ms).

## S3 Performance Analysis

In this section, we perform ablation study and analyze the impact of prior and generation network on the prediction performance. Specifically, we evaluate the prediction performance of CNN and the generation network (Figure S1) separately to see which part of the network pathway is more important in getting high accuracy. We summarize the results in Table S5.

| Model | CUB (val) | | | | | | LFW (val) | | |
|---|---|---|---|---|---|---|---|---|---|
| | pixel | | | IoU | | | pixel | | |
| | CNN | GEN | BOTH | CNN | GEN | BOTH | CNN | GEN | BOTH |
| CNN (baseline) | 91.17 | – | – | 79.64 | – | – | 92.09 | – | – |
| GDNN | 89.28 | 90.95 | 92.92 | 76.07 | 79.01 | 83.20 | 91.94 | 87.14 | 93.59 |
| GSNN | 86.19 | 90.95 | 93.09 | 68.61 | 79.38 | 83.62 | 92.08 | 87.95 | 93.71 |
| CVAE | 77.06 | 92.66 | 92.72 | 46.45 | 82.80 | 82.90 | 70.14 | 92.43 | 93.29 |
| hybrid | 89.42 | 91.65 | 93.05 | 75.04 | 80.93 | 83.49 | 91.74 | 92.80 | 93.69 |

Table S5: Prediction performance results on CUB and LFW validation sets. The columns with "CNN" refer the performance using the CNN output only, those with "GEN" refer the performance using the generation network output only, and those with "BOTH" refer the performance using the sum of outputs from both networks (which is the final model output evaluated in the main text).

Interestingly, the performance of the generation network output is higher than that of CNN in many cases. This behavior becomes more significant for CVAE, and the performance of the generation network output is already as good as the performance using both networks.