[Reviews · NeurIPS 2015]

Submitted by Assigned_Reviewer_1

Summary: A framework for learning complex structured output representations is presented. To this end variational auto-encoders (VAE) are extended to `conditional VAEs,' i.e., conditioned on the input data x.

Quality - The paper is mostly well written, could however be improved occasionally. Clarity - The idea is clearly presented but some details are missing. Originality - Conditional VAEs seem to be a straightforward extension of standard VAEs, but certainly worth a discussion Significance - The significance could be improved by a more extensive evaluation showing results for various modifications

Comments: - I think the term generative is typically used when learning distributions that also involve the input data. This seems different from the proposed conditional variational auto-encoder technique that is not capable of generating data via p(x). Hence readers could get confused.

- Three different methods are possible for inference, i.e., estimation of y given x. Two of them are evaluated using toy data. But which of the ones discussed in Sec. 3.1. do the authors recommend?

- The authors mention a gap (l. 148) measurable by the regularization term during/after learning, i.e., the proposal q(z|x,y) and the prior p(z|x) don't exactly agree. How big is this gap?

- The proposed solution to close the gap is to introduce a Gaussian stochastic neural network which models the reconstruction term directly, i.e., no regularization using a KL divergence. A weighted combination of the conditional VAE (computes regularization+reconstruction) and the newly introduced network (computes reconstruction) is then proposed as the desirable cost function. This however seems counter-intuitive to me. I would have expected the regularization gap between both distributions to be closer to zero if more weight is placed on the regularization rather than on the reconstruction term. Could the authors provide intuition? How did the authors set/cross-validate \alpha and what was the resulting value?

- There is a significant amount of semantic image segmentation techniques using CNNs. For fairness the authors might consider citing a few more recent ones.

- More details regarding the noise injection into data x is useful for a reader. What exactly did the authors do?

- Moreover as a reader I'm very interested in quantitative results regarding the modifications described in Sec. 3.3.2, i.e., how important is the `latent-to-output' pathway really? In addition I'm curious as to how much performance improved by using the direct output prediction as the input for the conditional prior network? Further I'd appreciate if the authors could clarify whether the conditional prior network as trained using only the direct output prediction as its input or whether both data x and prediction \tilde{y} were used? Although some of the modifications are investigated in Tab. 3, a more careful ablation analysis would seems very useful to a reader.

- Since the conditional VAE involves multiple networks I'm wondering whether its improved performance is a result of the larger amount of trainable parameters. Can the authors comment?

- Since efficiency is claimed in the abstract I'm wondering about the time for training and inference. Can the authors comment?

------------- As pointed out by fellow reviewers some citations could be added for completeness.
Summary: Extending variational autoencoders to conditional distributions seems valuable. A more extensive evaluation and including missing details could improve the paper.

Submitted by Assigned_Reviewer_2

This paper extends variational auto-encoders to the structured prediction setting, where now it is used to model the probability of the structured output given the input. It adds several innovations to the training process, and shows results on pixel labeling on CUB and LFW. The paper is novel to the best of my knowledge, and using such expressive models for structured prediction is I think an important goal and something that can have a high impact. The aims of this paper and the ideas presented in this paper are therefore very welcome. However the paper falls short in two ways: - The paper is not very clearly written. It is missing several details. First, what is the architecture of all the nets? The authors mention at some point that all networks are MLPs with 2 layers, but that can't be the full story because an MLP is not a generative model. Perhaps the authors mean that the MLP predicts the parameters of a Gaussian distribution as in [24] or a mixture model in [16]? It is also not clear how p(y|x,z) is being modeled by two networks net_{z2y} and net_{x2y}. Do these two networks produce predictions that are averaged together? Multiplied together? Later researchers trying to reproduce results will have a hard time. - The results are a bit underwhelming. Tables 3 and 4 show that the CNN is remarkably competitive while also being much simpler. Is the 1 point improvement worth it? (I am also not sure of what a 1 point gain in these datasets means). Table 5 does show good results, but the setting is a bit contrived.

In total, I like the ideas described in this paper and believe they deserve exploration. However, the results are not very great, and the paper requires some rewriting to make it clearer.
Summary: This paper uses recent progress in variational autoencoders to help structured prediction. While the aims are noble, the results are slightly underwhelming, and several details are missing.

Submitted by Assigned_Reviewer_3

In particular, the prediction of a structured output that is not the input data (ex. predicting segmentation labels) starts to make the use of the terminology of auto-encoders a bit of a stretch. Predicting the other half of an image stays within the normal conceptual framework of an autoencoding scheme.

The use of the term 'conditional *prior*' deviates a little far from what I think some members of the NIPS community would consider as appropriate and acceptable terminology.

Why not just call P(z|x) the conditional distribution of the hidden variable or the stochastic representation z or something else. Re-defining the concept of a prior as a 'conditional prior' could be quite incompatible with some already very established terminology about what it means to be a 'prior'. I very much understand the motivations here; however, I feel the terminology should be more precise in a publication.

Please give some more details on the baseline CNN, details can be important to understand if this is a strong or straw-man baseline. There are also a number of minor language errors throughout the manuscript that need to be fixed.
Summary: This paper explores a natural variation of stochastic conditional models, presented here as a conditional variational auto-encoder. The essence of the idea here is reasonable, and the experimental work is quite extensive. However, some terminology used in the paper stretches the accepted interpretations of certain concepts a little too far (see more detailed comments for specifics).

Submitted by Assigned_Reviewer_4

This paper describes a generative model for structured output variables by extending the variational auto-encoder to a conditional model . The authors propose a few different models and demonstrate on a pixel labelling tasks on toy MNIST data as well as real world CUB and LFW data, showing good results.

Aside from a few typos the paper is well written and clear. It is however slightly unclear how a final pixel labelling is produced, so perhaps a quick explanation illustrating the use of section 3.1 for pixel labelling would be beneficial to readers. There are some very related missing references: [Deep Structured Output Learning for Unconstrained Text Recognition, ICLR 2015] and [Learning Deep Structured Models, ICLR 2015].
Summary: Interesting paper on conditional variational autoencoder framework for pixel labelling. Other structured prediction tasks would have been interesting to see as well as a clear expose of the inference process.

Author Feedback
Author rebuttal: Thank you for the valuable comments. We will reflect all your comments and revise to clarify terminologies, provide insights via ablation study, and cite recent works on semantic segmentation and structured prediction for final version. We will release the code for reproducible research.

R1,R2,R6: [net architecture]
The baseline CNN consists of two nets, p(z|x) (net_x2z) and p(y|x,z) (net_x2y, net_z2y). The condVAE has recognition model q(z|x,y) (net_xy2z). We provide net architecture in Sec. 5.2:

net_x2z,net_xy2z: conv(9,64,2)-pool(2)-conv(5,96)-pool(2)-conv(3,128)-pool(2)-conv(3,32)
net_z2y: conv(3,96)-unpool(2)-conv(5,64)-unpool(2)-conv(5,48)-unpool(2)-conv(5,48)-unpool(2)-conv(5,2)
net_x2y: conv(9,64,2)-pool(2)-conv(5,96)-pool(2)-conv(3,128)-conv(3,128)-conv(3,96)-unpool(2)-conv(5,64)-unpool(2)-conv(5,48)-unpool(2)-conv(9,2)

* net_xy2z has similar architecture to net_x2z (# of input channels may differ depending on whether we use flat or recurrent prediction net); parameters are not shared.
* conv(kernel size, # kernels, stride (default=1)).
* (un)pool(pooling size): max pooling, stride=same as pooling size.
* we use ReLU after each intermediate convolution operation.

We draw z~net_x2z(x) and predict y using softmax(net_x2y(x)+net_z2y(z)). For recurrent prediction net, we draw z~net_x2z(x,\tilde{y}), \tilde{y}=softmax(net_x2y(x)).

R1,R3: [efficiency]
We compare mean processing time (in ms) per image (128x128) for training (on GeForce GTX TITAN):
CNN: 7.7ms (inference), 17.3ms (backprop)
condVAE: 10.3ms (inference), 31.3ms (backprop)

For testing, inference time should be the same as CNN since recognition model is not used. Computation increases for iterative inference (Eq 11), but increment is not linear to # of iteration since we can save some after initial computation (e.g., 49.3 ms for 10 iter).

R1: [analysis]
Thanks for detailed comments. We will include more ablative analysis.

1) we evaluate the prediction performance of each pathway on CUB (fold1). Considering overall performance (92.83%), the latent-to-output pathway (92.77%) turns out to be much more important than the input-to-output pathway (i.e., \tilde{y}; 81.30%) in condVAE. Here we didn't pretrain the input-to-output pathway, and we will include results for the case when the input-to-output pathway is pre-trained.

2) we used both x and \tilde{y} as input to net_x2z. In control experiments, we tried using \tilde{y} as a sole input, but preliminary results were worse than using both.

3) we conducted an experiment on MNIST ("2 quad" case) while varying # of trainable parameters. We found that even with smaller net, the condVAE showed higher validation CLL (# trainable parameters are shown in the parenthesis):

Table 1: [784-1000-1000-200-1000-1000-784]: -44.96 (condVAE; 896K), -62.14 (CNN; 638K)
condVAE with reduced parameters: [784-500-500-200-500-500-784]: -46.83 (323K)

R1: [hybrid objective]
In condVAE training, two terms in Eq 6 interact via recognition model: i.e., predicting z from q(z|x,y), where z is used to predict y. In testing, recognition model is not used. As you suggested, one way to resolve incompatibility in inference during training and testing is to emphasize the KL term more to minimize the gap between q(z|x,y) and p(z|x) (i.e., this has an implicit effect of making p(z|x) being more predictive of y). In control experiments, the preliminary results show that larger weight for the KL term gives slightly better prediction performance.

Instead, in our proposal, we train generation network (net_z2y) to be compatible to both training and testing by combining with GSNN objective (note that since GSNN uses p(z|x) for inference in both training and testing, so decreasing alpha in Eq 10 has an effect of making p(z|x) being more predictive of y). In sum, we believe these approaches are indeed complementary, and we will explore the extension of hybrid objective that controls both KL term and GSNN objective.

R1: [noise injection, sampling]
We generate a squared mask at random location of input image and set pixel values 0. The mask width is sampled less than 40% of the original width. We note that this can be viewed as a dropout in data layer using structured mask.

We recommend importance sampling as it requires much less samples (~100) than generative sampling (>10,000).

R5: [Comparison to DBN]
Our method has advantages over DBN that makes it suitable for structured prediction. First, our model is end-to-end trainable. In contrast, the DBN involves intractable partition function (of the top-layer RBM) which makes end-to-end training extremely difficult. In addition, our method doesn't require unsupervised pre-training, whereas the greedy training of DBN is "unsupervised" and doesn't necessarily lead to a good initialization for structured prediction. In addition, our model is scalable in network depth and the number of output variables using convolution and pooling.